

# Excitations and ergodicity of critical quantum spin chains from non-equilibrium classical dynamics

Stéphane Vinet[1], Gabriel Longpré[1] and William Witczak-Krempa[1,2,3]

**1** Département de Physique, Université de Montréal, Montréal, Québec, H3C 3J7, Canada
**2** Centre de Recherches Mathématiques, Université de Montréal; P.O. Box 6128, Centre-ville Station; Montréal (Québec), H3C 3J7, Canada
**3** Regroupement Québécois sur les Matériaux de Pointe (RQMP)

## Abstract

We study a quantum spin-1/2 chain that is dual to the canonical problem of non-equilibrium Kawasaki dynamics of a classical Ising chain coupled to a thermal bath. The Hamiltonian is obtained for the general disordered case with non-uniform Ising couplings. The quantum spin chain (dubbed Ising-Kawasaki) is stoquastic, and depends on the Ising couplings normalized by the bath's temperature. We give its exact ground states. Proceeding with uniform couplings, we study the one- and two-magnon excitations. Solutions for the latter are derived via a Bethe Ansatz scheme. In the antiferromagnetic regime, the two-magnon branch states show intricate behavior, especially regarding their hybridization with the continuum. We find that that the gapless chain hosts multiple dynamics at low energy as seen through the presence of multiple dynamical critical exponents. Finally, we analyze the full energy level spacing distribution as a function of the Ising coupling. We conclude that the system is non-integrable for generic parameters, or equivalently, that the corresponding non-equilibrium classical dynamics are ergodic.

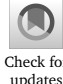

# 1  Introduction

The exact correspondance between the non-equilibrium dynamics of certain classical systems and the dynamics of closed quantum systems has a rich history [1]. This classical-quantum duality follows from the fact that the Markov or Master equation can be mapped to a quantum Schrödinger equation in imaginary time. One can thus gain insights about the dynamics from either side of the correspondance. This has been used in the study of various systems, such as the non-equilibrium dynamics of particles or spins on the lattice, and, on the quantum side of the mapping, various quantum spin Hamiltonians including Heisenberg and XY models, as well as dimer models [1–7]. One powerful application has been to use the integrability of certain quantum spin chains to solve the corresponding non-equilibrium classical dynamics [1]. Alternatively, one can generate new quantum Hamiltonians starting from classical dynamical models. This is the approach that we shall take in this work.

Our analysis begins with a canonical problem in non-equilibrium statistical mechanics: the non-equilibrium dynamics of a 1d classical Ising spin chain subject to Kawasaki (spin-preserving) dynamics. More precisely, the Ising chain couples to a thermal bath that allows anti-aligned neighboring spins to exchange positions. The corresponding quantum Hamiltonian is a spin-1/2 chain with a global U(1) symmetry, and contains up to 4-spin interactions. We shall refer to it as the *quantum Ising-Kawasaki chain*. In the limit of uniform Ising couplings (we also discuss the non uniform case), the quantum spin chain depends on a single paramater: $\beta J$, the Ising coupling normalized by the temperature of the classical bath, $1/\beta$. This temperature does not correspond to a temperature in the quantum system, instead it simply determines the coupling of the quantum spin chain. The latter is studied as a closed quantum system evolving under unitary Hamiltonian evolution. The exact groundstates are simply obtained from the Boltzmann distribution of the classical Ising model; the ground-state degeneracy is $N + 1$ corresponding to the different magnetization sectors for a chain of $N$ sites. The 1-magnon excitations (where 1 spin is flipped in a ferromagnetic sea) are obtained exactly, and disperse quadratically at low wavectors. Combined with the analysis from previous works [8–15], this leads us to conclude that the quantum spin chain, and corresponding non-equilibrium dynamics, host multiple dynamical critical exponents, one of them being $z = 2$. Next, we derive solutions for the two-magnon spectra via a Bethe Ansatz scheme. In the antiferromagnetic regime, the 2-magnon branch states show intricate behavior, especially regarding hybridization with the continuum. Next, we turn to the study of the entire eigen-spectrum, and numerically analyze the energy level statistics using exact diagonalization on short chains, $N \leq 25$. We conclude that the system is non-integrable for generic values of $\beta J$, or equivalently, that the corresponding non-equilibrium classical dynamics are ergodic as

expected.

Interestingly, the quantum Ising-Kawasaki chain shows phenomena that are similar to certain frustration free quantum spin chains that were recently introduced with motivation from a quantum information perspective: the Motzkin [16,17] and Fredkin [18] spin chains. Just like the Ising-Kawasaki chain, these gapless chains show multiple dynamics at low energy [19,20]. Besides the rich dynamical and entanglement properties of these chains, it has been shown that the Motzkin chain realizes a local approximate quantum error correcting code in its ground space [21]. It would be interesting to study the Kawasaki chain from such a quantum information perspective.

The rest of the paper is organized as follows. In Section 2 we introduce the classical non-equilibrium model, and the resulting quantum Hamiltonian. We give its exact groundstates. In Section 3, turning to the uniform case, we find the 1- and 2-magnon excitations via the standard Bethe Ansatz. Among others, we show exactly that the dynamical critical exponent in those sectors is $z = 2$. In Section 4, we use exact diagonalization to determine the energy level spacing distribution for the entire spectrum. Our analysis shows that the system is non-integrable for generic $\beta J$, which in turn implies that the classical Kawasaki dynamics are ergodic. We give a summary of our results and various extensions in Section 5.

## 2 The Model

As the Ising model is a static model in equilibrium, dynamical generalizations have been introduced notably by Glauber and Kawasaki [22,23] in order to study relaxational processes near equilibrium. Glauber dynamics consists of single spin-flip processes while Kawasaki proposes a spin exchange for a pair of unequal spins. Kawasaki dynamics thus conserves the total magnetization. The quantum Kawasaki spin chain is obtained from the corresponding Markov operator [5], as we now discuss.

For a classical Ising chain of length $N$, a state is identified by a vector $\vec{s} \in \{-1, 1\}^N$. We generalize the analysis of [2, 5] by working with non uniform couplings, for which the Ising energy reads

$$E_I(\vec{s}) = -\sum_{i=1}^{N} J_i s_i s_{i+1}, \tag{1}$$

where the coupling constant $J_i$ is positive/negative for ferromagnetic/antiferromagnetic interactions. Assuming periodic boundary conditions, we have that $s_{N+1} = s_1$. In thermal equilibrium, the probability to find a state is given by its Boltzmann weight $P(\vec{s}) = e^{-\beta E_I(\vec{s})}/Z$, where $\beta = 1/(k_B T)$ is the inverse temperature of the heat bath and $Z$ is the canonical partition function. After endowing the system with Kawasaki *non-equilibrium* dynamics, the evolution of the probability obeys the gain-loss equation:

$$\partial_t P(\vec{s}, t) = \sum_{\vec{s}' \in \{-1,1\}^N} \left[ W(\vec{s}' \to \vec{s}) P(\vec{s}', t) - W(\vec{s} \to \vec{s}') P(\vec{s}, t) \right], \tag{2}$$

which describes the evolution of the probability $P(\vec{s}, t)$ that the system will be in the state $\vec{s}$ at time $t$. In order to derive all subsequent probability distributions from the action of an evolution operator on a given initial distribution, Eq. (2) can be viewed as the Schrödinger evolution in imaginary time of a pseudo-Hamiltonian such that $\partial_t |P(t)\rangle = -\hat{W} |P(t)\rangle$. Consequently, $|P(t)\rangle = e^{-\hat{W}t} |P(0)\rangle$, where $\hat{W} = \hat{W}_d + \hat{W}_{nd}$ whose diagonal and non-diagonal matrix entries are

$$\langle \vec{s}| \hat{W}_d |\vec{s}\rangle = \sum_{\vec{s}' \neq \vec{s}} W(\vec{s}' \to \vec{s}) \quad \text{and} \quad \langle \vec{s}'| \hat{W}_{nd} |\vec{s}\rangle = -W(\vec{s} \to \vec{s}'). \tag{3}$$

The dimensionless transition probability rate for a state configuration $|\vec{s}\rangle$ to evolve to $|\vec{s}'\rangle$ is taken to be

$$W(\vec{s} \to \vec{s}') = \frac{1}{2}\left[1 - \tanh\left(\frac{\beta}{2}\Delta E_{\vec{s},\vec{s}'}\right)\right], \tag{4}$$

where $|\vec{s}\rangle$ and $|\vec{s}'\rangle$ differ at most by a pair of nearest neighbor spins (NN), and $\Delta E_{\vec{s},\vec{s}'} = E_I(\vec{s}') - E_I(\vec{s})$ is the Ising energy difference between the two configurations [23]. Using the $z$-axis in spin space as the quantization axis by promoting $s_i$ variables to Pauli operators $\sigma_i^z$, the obtained Markov operator $\hat{W}$ is real but not symmetric. The non-unitary similarity transformation $H' = M\hat{W}M^{-1}$, with

$$M = \exp\left(-\frac{1}{2}\sum_{i=1}^{N}\kappa_i\sigma_i^z\sigma_{i+1}^z\right), \tag{5}$$

where $\kappa_i = \beta J_i$, yields a self-adjoint Hamiltonian operator $H'$. It is important to note that $H'$ and the Markov operator $\hat{W}$ share the same eigenvalues. In addition, an eigenstate $|\psi\rangle$ of $H'$ yields an eigenstate $M|\psi\rangle$ of $\hat{W}$. Under periodic boundary conditions (PBCs), we find that $H'$ takes the following form:

$$\begin{aligned}
H' = &-\sum_{i=1}^{N}\frac{(\cosh(\kappa_{i+1})\cosh(\kappa_{i-1}) + \sigma_{i-1}^z\sigma_{i+2}^z\sinh(\kappa_{i+1})\sinh(\kappa_{i-1}))}{2(\cosh(2\kappa_{i+1}) + \cosh(2\kappa_{i-1}))}(\sigma_i^x\sigma_{i+1}^x + \sigma_i^y\sigma_{i+1}^y) \\
&+ \frac{1}{8}\sum_{i=1}^{N}[2 + (-2 - \tanh(\kappa_i + \kappa_{i-2}) - \tanh(\kappa_i - \kappa_{i-2}) - \tanh(\kappa_{i+2} + \kappa_i) \\
&\quad + \tanh(\kappa_{i+2} - \kappa_i))\sigma_i^z\sigma_{i+1}^z + (\tanh(\kappa_{i+1} + \kappa_{i-1}) + \tanh(\kappa_{i+1} - \kappa_{i-1}) \\
&\quad + \tanh(\kappa_{i+2} + \kappa_i) - \tanh(\kappa_{i+2} - \kappa_i))\sigma_i^z\sigma_{i+2}^z],
\end{aligned} \tag{6}$$

where the first line generates spin-flips, while the last are diagonal in the $\sigma_i^z$ basis. We note that in the limit of infinite temperature, $\kappa_i = 0$, we recover the *uniform* ferromagnetic Heisenberg XXX Hamiltonian. This is because the individual values of the couplings $J_i$ become unimportant compared to the large bath temperature $1/\beta$. In that limit, the dynamics simplify, and in fact become integrable. We shall see that this is not the case at finite $\beta$.

The Hamiltonian (6) is *stoquastic* in the standard $\sigma_i^z$ basis. A Hamiltonian is stoquastic with respect to a given basis if it has only real nonpositive off-diagonal matrix elements in that basis [24]. This can be readily seen from the non-diagonal term of Eq. (6) as its matrix elements $-\operatorname{sech}(\kappa_{i+1} \pm \kappa_{i-1})/2$ are real and non-positive $\forall \kappa_i$. This implies that its ground states can be expressed as a classical probability distribution [25]. Furthermore, stoquastic Hamiltonians avoid the "sign problem" in quantum Monte Carlo (QMC) algorithms. In particular, the standard quantum-to-classical mapping used in QMC algorithms does not result in a partition function with Boltzmann weights taking negative or even complex values [26].

For uniform couplings, we find that Eq. (6) reduces to:

$$H = \gamma_\kappa\sum_{i=1}^{N}\left(1 + \delta_\kappa\sigma_{i-1}^z\sigma_{i+2}^z\right)\left(\sigma_i^x\sigma_{i+1}^x + \sigma_i^y\sigma_{i+1}^y\right) + \frac{1}{4}\sum_{i=1}^{N}\left[1 + \alpha_\kappa\sigma_i^z\sigma_{i+2}^z - (1 + \alpha_\kappa)\sigma_i^z\sigma_{i+1}^z\right], \tag{7}$$

where $\gamma_\kappa = -\frac{1}{8}(1 + \operatorname{sech}(2\kappa))$, $\delta_\kappa = \tanh(\kappa)^2$, $\alpha_\kappa = \tanh(2\kappa)$, in agreement with [5].

At $\kappa = \infty$, we obtain $\gamma_\kappa = -1/8$ and $\delta_\kappa = \alpha_\kappa = 1$. The Hamiltonian then takes the form of the folded spin-1/2 XXZ model with an extra diagonal term, as seen for example in [27].

## 2.1 Ground states

As the total spin is conserved in the interaction with the thermal bath, the resulting Hamiltonian (7) conserves total $S_T^z$ (the spin symmetry is enlarged to SU(2) at $\beta = \infty$). We thus

have a thermodynamic equilibrium for every spin subsector. The $N + 1$ ground states of the Hamiltonian (7) are thus obtained by applying the nonunitary similarity transformation (5) to Ising equilibrium states in each of these subsectors.

At thermodynamic equilibrium, the probability of finding the Ising chain in a configuration $\vec{s}$ is given by the Boltzmann distribution

$$\text{Prob}_{\text{eq}}(\vec{s}) = \frac{1}{Z} \exp(-\beta E_I(\vec{s})) = \frac{1}{Z} \exp\left(\sum_{i=1}^{N} \kappa_i s_i s_{i+1}\right). \tag{8}$$

Suppose the total spin $S_T^z = j$ for $j \in \{-N/2, -N/2 + 1, \ldots, N/2 - 1, N/2\}$, the Ising thermal equilibrium state for the subsector $\mathcal{S}_j$ can be mapped to the following (not normalized) quantum state

$$|P_{\text{eq},j}\rangle := \sum_{\vec{s} \in \mathcal{S}_j} \text{Prob}_{\text{eq}}(\vec{s}) |\vec{s}\rangle = \frac{1}{Z} \sum_{\vec{s} \in \mathcal{S}_j} \exp(-\beta E_I(\vec{s})) |\vec{s}\rangle, \tag{9}$$

where $|\vec{s}\rangle$ is labelled by the $\sigma_i^z$ eigenvalues. The ground state is then obtained by applying the transformation (5) to $|P_{\text{eq},j}\rangle$, and normalizing the result. Therefore the ground state of $H$ for the subsector of total spin $S_T^z = j$ is

$$|\phi_{0,j}\rangle = \frac{M |P_{\text{eq},j}\rangle}{\langle P_{\text{eq},j} | MM | P_{\text{eq},j}\rangle^{1/2}} = \frac{1}{Z_j^{1/2}} \sum_{\vec{s} \in \mathcal{S}_j} \exp\left(-\frac{\beta}{2} E_I(\vec{s})\right) |\vec{s}\rangle, \tag{10}$$

where $Z_j = \sum_{\vec{s} \in \mathcal{S}_j} \exp(-\beta E_I(\vec{s}))$ is the partition function restricted to the magnetization sector $j$. Up to an overall constant, each component or amplitude of the groundstate vector (10) is the square root of the amplitude appearing in the original equilibrium state $|P_{\text{eq},j}\rangle$.

We note that the ground state degeneracy is no longer $N + 1$ at $\beta = \infty$, i.e. as the bath temperature vanishes. The probability rate in Eq. (4) is zero for transitions that would raise the Ising energy ($\Delta E_{\vec{s},\vec{s}'} > 0$). Consequently, the classical system will often get stuck in one of the many metastable states for which the only possible transitions would increase the energy. These states correspond to configurations in which each spin kink (domain wall) is separated by more than a nearest-neighbor distance from any other spin kink [8]. The quantum system is further constrained at low temperature by the non-unitary transformation $M$, which suppresses the energy lowering transitions, and leads to the conservation of the Ising energy $H_{\text{Ising}} = \sum_i \sigma_i^z \sigma_{i+1}^z$, which is directly related to the domain-wall number $n_{\text{dw}} = \frac{1}{2}(N - H_{\text{Ising}})$. This is shown in Appendix A. These additional constraints at $\beta = \infty$ give rise to glassy dynamics. We find numerically for the quantum spin chain up to $N = 24$ that the number of ground states $g(N)$ of $H$ at $\kappa = \infty$ is exactly given by

$$g(N) = \left(\frac{1 + \sqrt{5}}{2}\right)^N + |(N - 1 \bmod 6) - 2| + \lfloor \frac{N-2}{3} \rfloor \bmod 2, \tag{11}$$

where $\frac{1+\sqrt{5}}{2}$ is the golden ratio, and $\lfloor x \rfloor$ corresponds to the floor, i.e. the greatest integer less than or equal to $x$. Given that Eq. (11) holds exactly for $N \leq 24$, we expect that it will hold for all $N$. Consequently, the ground state degeneracy grows exponentially as $(\frac{1+\sqrt{5}}{2})^N$ at large $N$, which is in accordance with the scaling of the number of stable configurations for the classical system [8, 28].

## 3 Spin wave solutions

In the remainder of the paper, we shall work with the case of uniform couplings $\kappa_i = \kappa$. The rotational symmetry about the $z$-axis in spin space implies that the total $z$-spin $S_T^z = \sum_{i=1}^{N} \sigma_i^z / 2$

is conserved. Consequently, the Hamiltonian matrix in Eq. (7) can be block diagonalized according to the total $z$-spin quantum number $S_T^z \in \{-N/2, -N/2+1, \ldots, N/2-1, N/2\}$. Each of its values corresponds to an eigenspace $\mathcal{S}_j$ of dimension $\binom{N}{N/2-S_T^z}$. Therefore, the subsectors with $S_T^z = \pm N/2$ consist of a single eigenstate. In section 3, we consider the one and two magnon sectors namely $S_T^z = N/2-1$ and $S_T^z = N/2-2$.

## 3.1 One-magnon sector

For the $S_T^z = N/2-1$ sector, the translational symmetry allows for the complete diagonalization of the Hamiltonian. Consequently, the eigenvectors of the translation operator

$$|\psi\rangle = \frac{1}{\sqrt{N}} \sum_{n=1}^{N} e^{ikn} |n\rangle \,, \tag{12}$$

where $|n\rangle$ is the state corresponding to the configuration in which the magnon is in the $n^{\text{th}} \in [1, N]$ site, with wavevector $k = 2\pi m/N$ and $m \in [0, N-1]$, are also eigenvectors of $H$ whose eigenvalues are given by

$$E_m = 1 - \cos(2\pi m/N)\,. \tag{13}$$

Setting $m = 0$, one recovers the corresponding ground state from the previous section. Eq. (12) is the well-studied single-magnon equation of the Heisenberg spin chain studied in [29, 30], among others, with a dynamical critical exponent of $z = 2$ and wavelength $\lambda = 2\pi/k$. Note that Eq. (13) is independent of $\kappa$ since the Ising energy does not change as the magnon hops. It can also be seen in the coordinate Bethe ansatz formalism as

$$|\psi\rangle = \sum_{n=1}^{N} a(n) |n\rangle\,, \tag{14}$$

where periodic boundary conditions impose $a(n) = a(n+N)$. The spin-flip component of the Hamiltonian applied onto the state $|n\rangle$ will yield a non-zero result when either $i = n$ or $i+1 = n$:

$$(\sigma_n^x \sigma_{n+1}^x + \sigma_n^y \sigma_{n+1}^y) |n\rangle = 2 |n+1\rangle \,, \tag{15a}$$

$$(\sigma_{n-1}^x \sigma_n^x + \sigma_{n-1}^y \sigma_n^y) |n\rangle = 2 |n-1\rangle \,, \tag{15b}$$

evaluating the remaining $\sigma^z$ terms in the eigenvalue equation $H|\psi\rangle = E|\psi\rangle$ one obtains

$$H \sum_{n=1}^{N} a(n) |n\rangle = \sum_{n=1}^{N} a(n) [2\gamma_\kappa (1+\delta_\kappa)(|n+1\rangle + |n-1\rangle) + |n\rangle]\,, \tag{16}$$

where $2\gamma_\kappa (1+\delta_\kappa) = -\frac{1}{4}(1+\text{sech}(2\kappa))(1+\tanh(\kappa)^2) = -\frac{1}{2}$ thus the coupling dependence cancels out and we find the following condition for the coefficients $a(n)$:

$$2Ea(n) = 2a(n) - a(n-1) - a(n+1)\,. \tag{17}$$

Consequently, in accordance with Eq. (12) we have that $a(n) = e^{ikn}$, as well as $N$ linearly independent solutions of Eq. (17) obeying periodic boundary conditions for $e^{ik(n+N)} = e^{ikn}$.

## 3.2 Two-magnon sector

Analogously for $S_T^z = N/2 - 2$, we have in the two magnon sector

$$|\psi\rangle = \sum_{1 \le n_1 < n_2 \le N}^{N} a(n_1, n_2)|n_1, n_2\rangle . \tag{18}$$

We work in the center of mass $K = k_1 + k_2$, with relative coordinate $j = n_2 - n_1$. Based on symmetry considerations, we use the following ansatz for the coefficients $a(n_1, n_2)$:

$$a(n_1, n_2) = e^{iK(n_1+n_2)} g(j), \tag{19}$$

where $K = m\pi/N$ is the momentum of the center of mass, $m = 0, 1, ..., N-1$ and $g(j)$ a function of the number of lattice sites between the two excitations. PBC impose that $a(n_1, n_2) = a(n_2, n_1 + N)$ and so $e^{iKN} g(N - j) = g(j)$. We consequently obtain a periodic condition dependent on the parity of $m$ as in the even case $g(j) = g(N-j)$ whereas in the odd case $g(j) = -g(N - j)$. For even $N$, these coefficients satisfy the following linear equations given by substituting Eq. (18) into the Schrödinger equation:

$$Eg(1) = (1 - \alpha_\kappa)g(1) + 4\gamma_\kappa(1 - \delta_\kappa)\cos(K)g(2), \tag{20}$$

$$Eg(2) = 4\gamma_\kappa(1 - \delta_\kappa)\cos(K)g(1) + (2 + \alpha_\kappa)g(2) - \cos(K)g(3), \tag{21}$$

$$Eg(j) = -\cos(K)[g(j-1) + g(j+1)] + 2g(j), \quad \text{for} \quad 3 \le j < N/2. \tag{22}$$

In the limiting case of Eq. (22) where the magnons are furthest apart $j = n_2 - n_1 = N/2$ we find that $g(j-1) = e^{im\pi}g(j+1)$ and so

$$Eg(N/2) = -2\cos(K)g(N/2 - 1) + 2g(N/2), \quad \text{for } m \text{ even}, \tag{23a}$$

$$Eg(N/2) = 2g(N/2), \quad \text{for } m \text{ odd}. \tag{23b}$$

For an odd spin chain length the upper bound of $j$ on Eq. (22) is reduced to $(N-1)/2$ and Eq. (23) is changed to

$$Eg\left(\frac{N-1}{2}\right) = -\cos(K)g\left(\frac{N-3}{2}\right) + (2 - \cos(K))g\left(\frac{N-1}{2}\right) \quad \text{for } m \text{ even}, \tag{24a}$$

$$Eg\left(\frac{N-1}{2}\right) = -\cos(K)g\left(\frac{N-3}{2}\right) + (2 + \cos(K))g\left(\frac{N-1}{2}\right) \quad \text{for } m \text{ odd}. \tag{24b}$$

One finds that Eq. (23b) is trivially satisfied by $E = 2$. This result would however yield an overcomplete set of nonorthogonal and nonstationary states. Removing the overcount, the remaining task is to solve the system of Eqs. (20)-(24) also known as the Bethe ansatz equations [31]. Note that we obtain particular equations for $g(1)$ and $g(2)$ corresponding to the nearest neighbor (NN) and next nearest neighbor (NNN) configurations which lead to distinct behavior and branch states in the two-magnon spectra. We explore this further in subsections 3.2.1 and 3.2.2 below. Additionally, we introduce the fidelity $F$ as

$$F(\text{NN}, \psi) = \frac{1}{N} |\langle \text{NN}|\psi\rangle|^2 = |g(1)|^2 , \tag{25}$$

where the NN state for a chain of $N$ sites is defined as

$$|\text{NN}\rangle = \frac{1}{\sqrt{N}} \sum_{n_1=1}^{N} e^{iK(n_1+n_2)} |\uparrow ... \uparrow\downarrow_{n_1}\downarrow_{n_2}\uparrow ... \uparrow\rangle , \tag{26}$$

where $n_2 = (n_1 + 1) \mod N$. It is defined as the probability to measure a given state $|\psi\rangle$ of momentum $K$ and energy $E$ in a bound state configuration corresponding to the two magnons being nearest neighbors. These bound state configurations were previously theoretically studied in [32–36] and from an experimental point of view as well in [37, 38].

### 3.2.1 Ferromagnetic $J > 0$

A plot of the two-magnon energies versus wavevector $K$ for $N = 200$ as obtained from the solutions of Eqs. (20)-(24) is shown in Fig. 1. The 19900 points in the range $0 \leq K \leq \pi$ produce a density plot for the two-magnon continuum which emerges in the limit $N \to \infty$ [30, 39]. The color scheme legend denoting the fidelity is also provided.

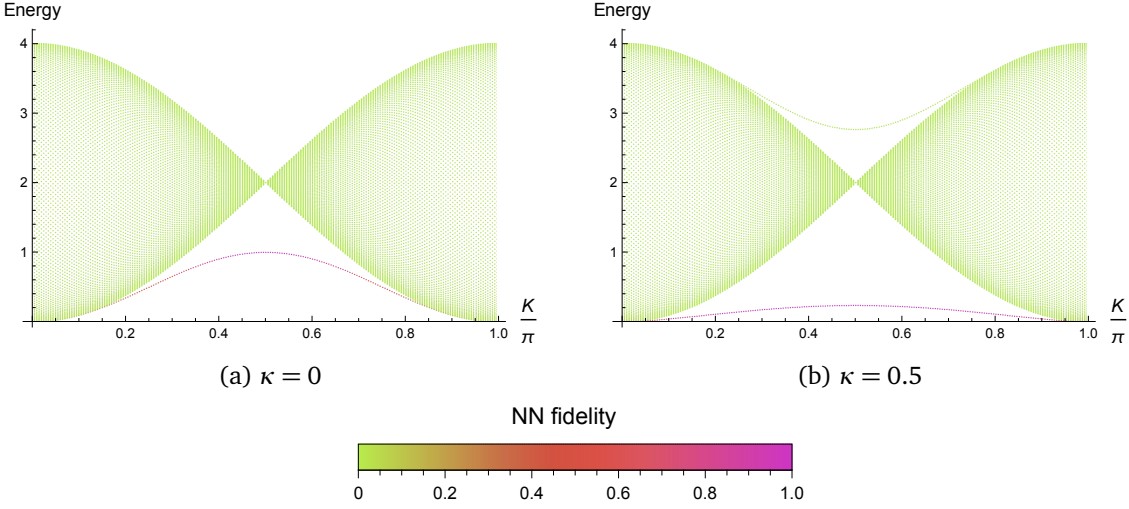

(a) $\kappa = 0$          (b) $\kappa = 0.5$

Figure 1: Two-magnon excitations and nearest neighbor fidelity of the ferromagnetic quantum Ising-Kawasaki spin chain with $N = 200$ for (a) $\kappa = 0$ and (b) $\kappa = 0.5$. For $\kappa = 0$ we recover the Heisenberg XXX known result [39].

We recover in Fig. 1a the Heisenberg XXX continuum when taking $\kappa = 0$ in accordance with Eq. (7). Generally, two-magnons configurations correspond to 4 domain walls. However, magnons on neighboring sites correspond to 2 domain walls and require intermediate states with a larger number of domain walls implying a higher energy in order to propagate. Thus nearest neighbor configurations have a lower energy and survive as a bound state. Consequently, as can be seen for the XXX Heisenberg model, pairs of magnons form a continuum of two-magnon scattering states and a lower branch of two magnon bound states [32,33,40,41]. Additionally, in Fig. 1b we find an upper branch of states separate from the continuum dominated by next-nearest neighbor excitations of the form $|\uparrow \ldots \uparrow\downarrow\uparrow\downarrow\uparrow \ldots \uparrow\rangle$. Consequently the lower and upper branches are respectively characterized by the Eqs. (20) and (21) for $g(1)$ and $g(2)$. The apex of each branch is found at $K = \pi/2$ as equations (20) and (21) are simplified to

$$(1 - \alpha_\kappa)g(1) = Eg(1), \tag{27}$$

$$(2 + \alpha_\kappa)g(2) = Eg(2). \tag{28}$$

The maximum of the lower branch of bound states decreases with increasing $\kappa$ in the ferromagnetic case, and in the limit $\kappa \to \infty$, the bound states tend towards zero. Meanwhile, the behavior of the upper branch at large $\kappa$ remains similar to what is seen in Fig. 1b, with its minimum increasing to $E = 3$. The two-magnon spectrum in this limit is shown in Fig. 8a of Appendix A.

### 3.2.2 Antiferromagnetic $J < 0$

In the antiferromagnetic regime $\kappa < 0$, the branch states have a more intricate behavior as we now discuss. At $\kappa = 0$, we saw previously that the branch below the continuum corresponds to

a bound state of the two magnons (high NN fidelity). As $\kappa$ becomes negative, anti-alignment of spins is favored, implying that this branch is pushed to higher energies. In addition, a NNN branch appears, as for $\kappa > 0$. These phenomena can be seen in Fig. 2a-2b. However, in contrast to the ferromagnetic case, the NNN is now located below the continuum. As the $\kappa$ becomes progressively more negative, the NNN branch shifts to lower energies, while the NN to higher energies. The inevitable collision between the two branches occurs at

$$\kappa_{\mathrm{x}} = \frac{\operatorname{arctanh}(-1/2)}{2} = -\frac{\ln 3}{4} \approx -0.275, \tag{29}$$

as can be observed in Fig. 2c. As $\kappa \to \kappa_{\mathrm{x}}$, the bound state (NN) fidelity of the NNN branch states increases. At $\kappa_{\mathrm{x}}$, we find that $(1-\alpha_\kappa) = (2+\alpha_\kappa)$, and so the energies found in Eqs. (27) and (28) are equal and the two branches cross at $K = \pi/2$. Additionally, the fidelity of the two branches becomes the same near the crossing, as can be seen in Fig. 2c.

In the limit where $\kappa \to -\infty$, the bound state fidelity of the NNN branch states decreases back to 0 past $\kappa_{\mathrm{x}}$. We also find that Eq. (20) becomes

$$2g(1) = Eg(1), \tag{30}$$

and the interaction term $4\gamma_\kappa(1-\delta_\kappa)\cos(K)$ between the coupled Eqs. (20) and (21) tends to zero, meaning that we have a pure bound state not interacting with the continuum of energy $E = 2$ and fidelity 1 at every value of $K$. This is shown in Fig. 8b of Appendix A. Consequently, the NN branch states form the upper branch of the continuum in the antiferromagnetic regime past $\kappa_{\mathrm{x}}$. This limit can be observed in Fig. 4.

We note that throughout Fig. 1 to 2 the bowtie form of the envelope of the continuum is unchanged implying that it is independent of $\kappa$. However, the states within the envelope evolve with $\kappa$. The coupling strength of the $g(1)$ dominated states, $|4\gamma_\kappa(1-\delta_\kappa)|$, decreases with increasing $|\kappa|$ and conversely increases as $|\kappa|$ decreases as can be seen by the black line in Fig. 4. For strong interactions, the quasiparticle is pushed outside of the continuum as a consequence of level repulsion [42]. This is observed in Fig. 2a. One would expect that as the interaction starts weakening, the bound state decays when encountering the continuum. It is indeed the case in Fig. 2d. However as the coupling further decreases the bound state becomes longer lived throughout the continuum as can be seen in Fig. 3 and Fig. 8b. In fact, the weaker interactions lead to a stronger NN fidelity and lower perturbation allowing for a penetration of the bound state branch through the continuum. Note that the NN fidelity observed in Fig. 2 is significantly lower due to a lower $|\kappa|$ leading to higher coupling in comparison with Fig. 3 and thus resulting in a faster decay of the branch states within the bulk of the continuum. This behaviour is particular to the antiferromagnetic regime as the bound state branch tends toward the center of the continuum at $E = 2$ with increasing fidelity in the $\kappa \to -\infty$ limit. In the ferromagnetic case, the bound state branch approaches $E = 0$ as $\kappa \to \infty$, so that the bound states remain below the continuum and no merger with the continuum occurs.

### 3.3 Dynamical critical exponents

The critical exponent of the 2-magnon sector can be easily deduced to be $z = 2$ from the quadratic shape of the enveloppe of the continuum, which holds for all values of the coupling $\kappa$. Note that this is the same critical exponent as obtained from Eq. (13) for the 1-magnon sector. This is also consistent with the critical exponent calculated by Grynberg for the entire spin chain with antiferromagnetic coupling $\kappa < 0$ in [5]. However, in the ferromagnetic case $\kappa > 0$, Grynberg numerically obtains a critical exponent dependent on $\kappa$, and always exceeding 2. In particular, it ranges from $z \approx 2$ at small $\kappa$, while in the limit where $\kappa \gg 1$ Grynberg finds a subdiffusive exponent $z \approx 3.1$ using exact diagonalization on short chains. For a range of

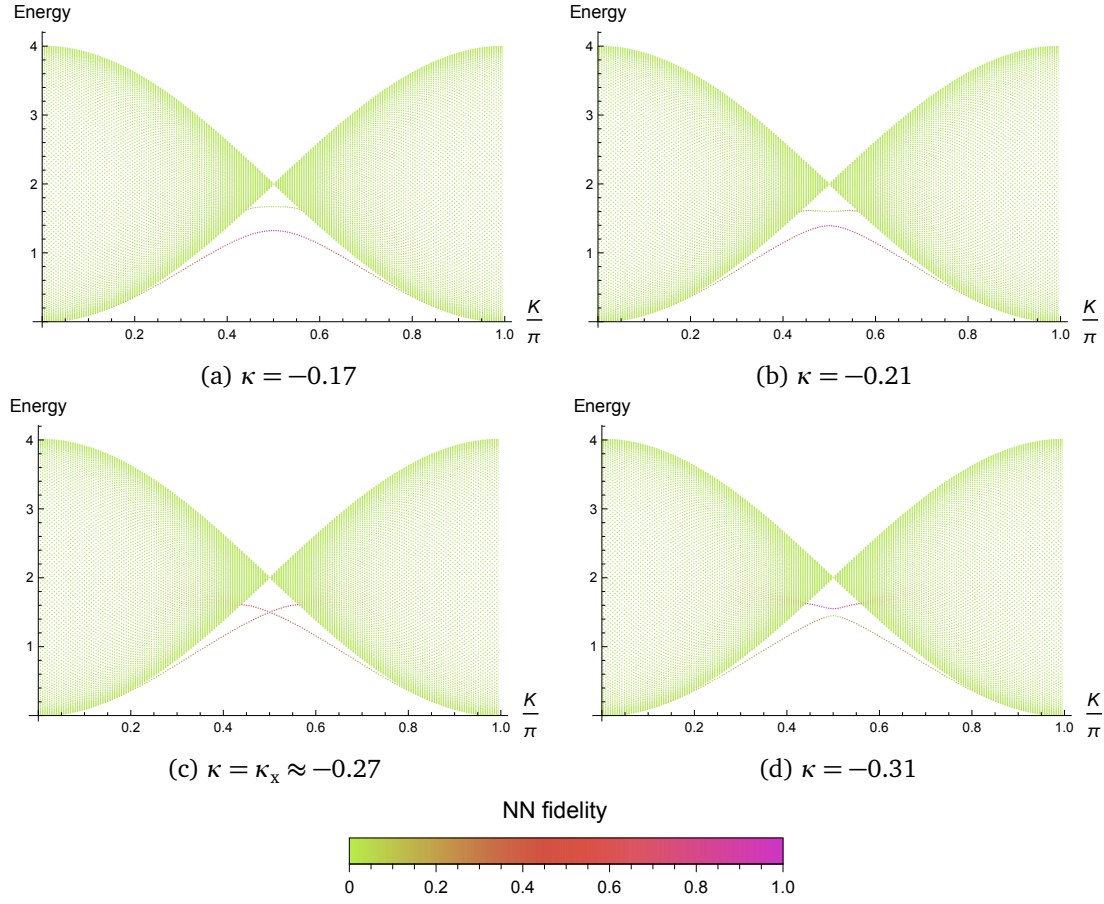

Figure 2: Two-magnon excitations and nearest neighbor fidelity for the antiferromagnetic quantum Ising-Kawasaki spin chain with $N = 200$ around $\kappa_x$.

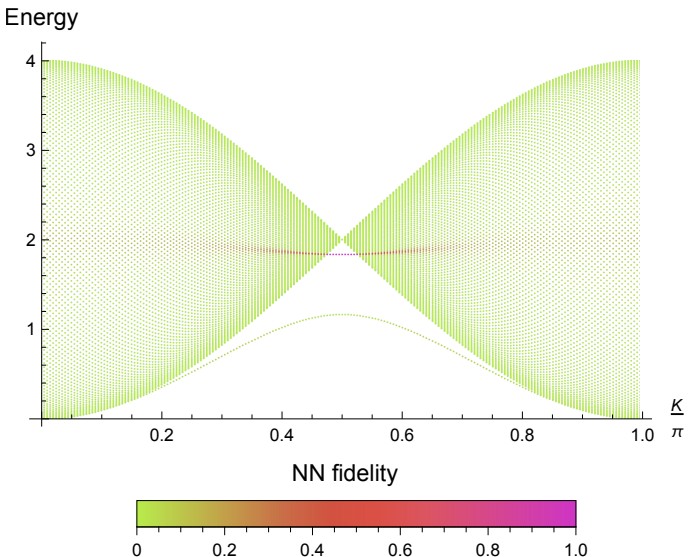

Figure 3: Two-magnon excitations and nearest neighbor fidelity for $N = 200$ and $\kappa = -0.6$.

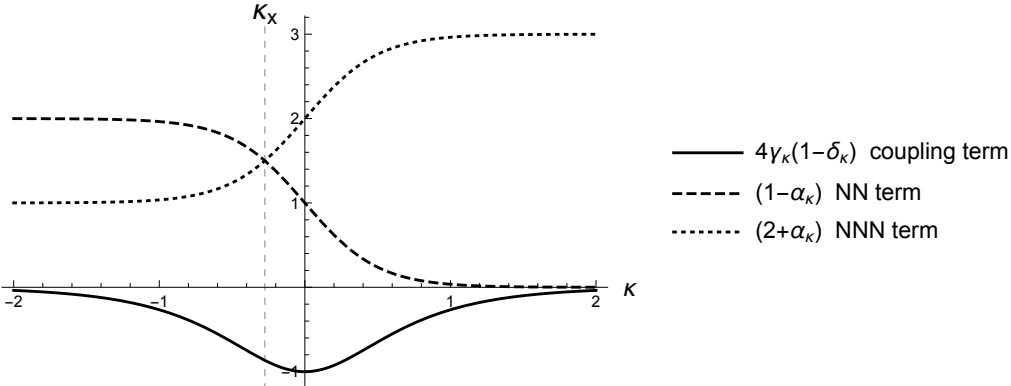

Figure 4: Nearest and next-nearest neighbor branch state energies for the two-magnon excitations at $K = \pi/2$ given by Eqs. (27,28). The crossing value where the Eqs. (27,28) have the same energy at $\kappa_x$ is highlighted by the grey vertical dashed line. The solid black line shows the coupling term in Eqs. (20, 21).

$\kappa > 0$ values, we have found using exact diagonalization that the lowest excited states occur in the $S_z^T = 0$ sector for even chains and the $S_z^T = \pm 1/2$ sectors for odd chains, which are substantially more complicated than the 1- or 2-magnon sectors. Furthermore, the momentum of these excited states corresponds to the smallest non-zero magnitude: $\pm 2\pi/N$. We thus see that the lowest excitated level has a degeneracy of 2 for even chains, while it is 4 for odd ones. We have also found that the next excited level has the same momenta, but higher spin: $S_z^T = \pm 1$ (even chains) and $S_z^T = \pm 3/2$ (odd chains).

Grynberg's results are consistent with the known results for classical Kawasaki dynamics at large $\beta$ according to which $z = 3$ is the dominant critical exponent in sectors with small $|S_T^z|$ [9–12]. We recall the basic argument here. When $\kappa$ is large but finite, the system rapidly reaches a metastable state via energy conserving and lowering events. Then after a long time $e^{4\kappa}$, an energy raising event overcomes the local barrier, and the system can reach a lower metastable state. Eventually, this cascade leads to the ground state. The key process is the diffusion of a spin accross a domain of opposite orientation, which can be alternatively seen as the slow diffusion of the entire domain [8, 13–15]. The diffusion coefficient scales as $D(\ell) = 1/\ell$ for a domain of length $\ell$, which leads to a subdiffusive domain growth $\ell \sim t^{1/3}$. This translates to a dynamical critical exponent $z = 3$ (relating energy to wavevector $E \sim k^z$). We note that in the classical statistical mechanics literature, the dynamical exponent is often defined as the inverse of the above definition: $z_c = 1/z = 1/3$. We thus see that the quantum spin chain hosts both diffusive ($z = 2$) and subdiffusive modes at $\kappa > 0$, signalling the presence of multiple dynamics at low energy. The subdiffusive modes soften as $\beta$ increases, and will thus dominate certain observables. Similar phenomena were encountered in the Motzkin and Fredkin quantum spin chains (and their deformations) [19, 20]. A dynamical critical exponent of $z = 3$ was also encountered in a different $S = 1/2$ quantum spin chain studied in the context of lattice supersymmetry [43, 44]. It would be desirable to obtain a low energy description that explains the critical exponents in such systems.

## 4 Energy Level Statistics

The level spacing distribution $P(s)$ is the probability that adjacent energies have spacing $s$. For the following analysis, we need to work with the unfolded and unsymmetrized spectrum. The unfolded spectrum is obtained by renormalizing the energies such that the local density of

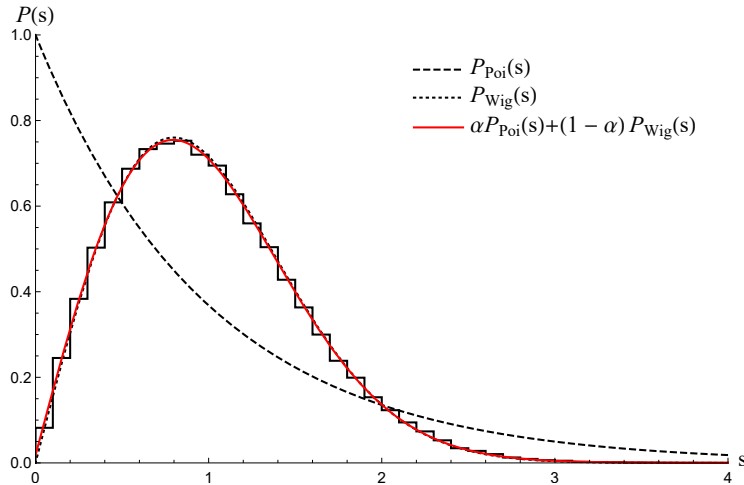

Figure 5: Level spacing distribution for $N = 24$, $\kappa = 0.5$ in the $S_T^z = 1$ and $k = 2\pi/24$ symmetry sector which has 104006 states. The dashed, dotted and plain red lines represent respectively the negative exponential distribution (31), the Wigner surmise (32), and the fit (33). The fit coefficient $\alpha$ value obtained is 0.02.

states is constant and equal to one. To do so, we first compute the full spectrum in a specified symmetry sector labeled by $S_T^z$ and $k$ as detailed in Appendices B and C. Then, the unfolding method presented in [45, 46] is followed to obtain the unfolded eigenvalues and their nearest neighbor level spacing $s$. We define the integrated density of states as $n(E) = \sum_{i=1}^{D} \theta(E - E_i)$, where $\theta$ is the Heaviside step function, $D$ is the dimension of the symmetry sector and $\{E_i\}_{i=1}^{D}$ the energies. We then approximate its average $\langle n(E) \rangle$ via a spline interpolation through the evenly spaced sample points $(E_i, n(E_i))$ for $i \in \{1, 21, 41, \dots\}$. The size of the sample step here chosen to be 20 doesn't affect the results when far enough from 1 and $D$. The unfolded eigenvalues are then defined as $x_i = \langle n(E_i) \rangle$ and their nearest neighbor spacing $s_i = x_{i+1} - x_i$ [45, 46]. According to the Berry-Tabor conjecture [47, 48], if the system is integrable, the distribution will be a negative exponential (31) describing a Poisson process:

$$P_{\text{Poi}}(s) = \exp(-s). \tag{31}$$

On the other hand, if it isn't integrable the spectrum will exhibit level repulsion and rigidity, and, for systems with time-reversal symmetry, the distribution will be the Wigner surmise (32) from the Gaussian orthogonal ensemble (GOE) of random matrix theory:

$$P_{\text{Wig}}(s) = \frac{\pi s}{2} \exp\left(-\frac{\pi s^2}{4}\right). \tag{32}$$

The distributions will be fitted with a normalized linear combination of the two distributions:

$$\alpha P_{\text{Poi}}(s) + (1 - \alpha) P_{\text{Wig}}(s). \tag{33}$$

Figure 5 shows the distribution for the largest symmetry sector with $S_T^z \neq 0$ and $k \notin \{0, \pi\}$ for a chain of length $N = 24$. Very strong correspondence with the Wigner distribution is observed.

In the $k \in \{0, \pi\}$ symmetry sectors, an extra desymmetrization step is needed to account for the spatial inversion symmetry $\hat{P}$. Figure 6 shows the distributions for both reflection eigenvalues $P = \pm 1$ in the largest symmetry sector with spatial reflection symmetry ($k = 0$) for $N = 25$. Again, Wigner distributions are observed.

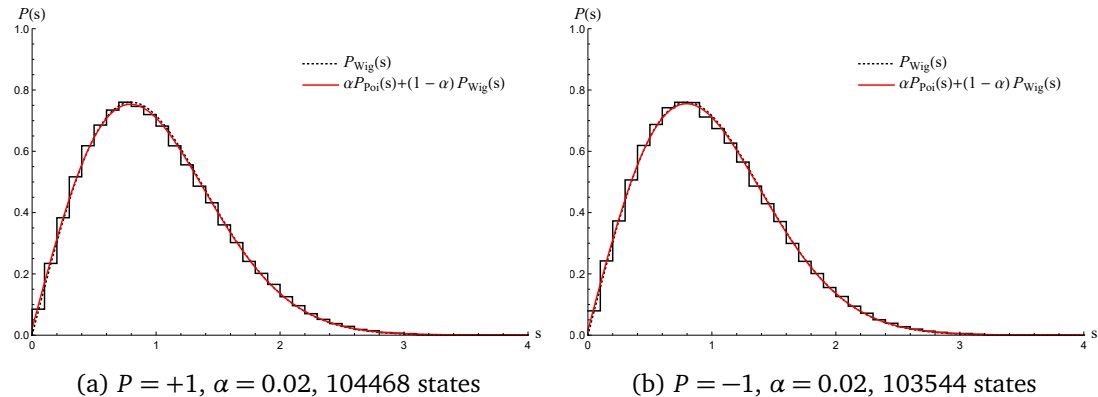

(a) $P = +1$, $\alpha = 0.02$, 104468 states $\qquad$ (b) $P = -1$, $\alpha = 0.02$, 103544 states

Figure 6: Level spacing distribution for $N = 25$, $\kappa = 0.5$ in the $S_T^z = 1/2$, $k = 0$ and $P = \pm 1$ symmetry sector.

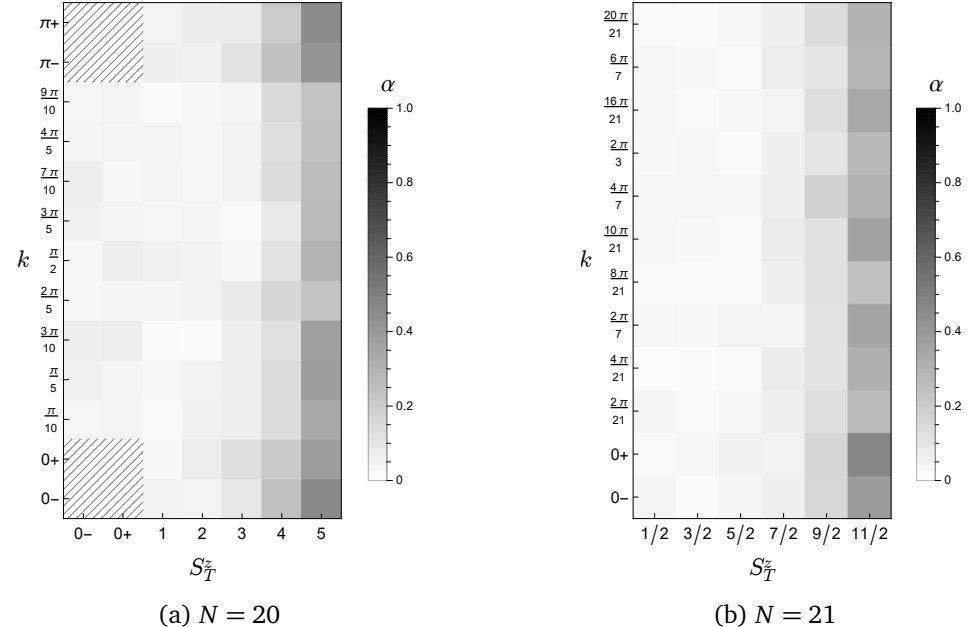

(a) $N = 20$ $\qquad\qquad\qquad$ (b) $N = 21$

Figure 7: Fit coefficient $\alpha$ in each symmetry sector for $N = 20$ and $N = 21$ with $\kappa = 0.5$. Values of $\alpha = 0$ and 1 correspond respectively to Wigner and Poisson distributions, respectively. The symbols $0\pm$ and $\pi\pm$ on the momentum axis $k$ designate the symmetry sectors with $P = \pm 1$ and respectively $k = 0$ and $\pi$. Similarly, the $0\pm$ on the total z-spin axis $S_T^z$ designate the symmetry sectors with $\mathcal{T} = \pm 1$ and $S_T^z = 0$. Here $\mathcal{T}$ corresponds to time-reversal. Sectors with $S_T^z \geq 6$ are not shown as they are too small ($< 300$) to produce a good histogram.

Spin reversal symmetry also implies that the spectrum for $S_T^z$ is the same as the one for $-S_T^z$. Similarly, the spatial inversion symmetry implies that the spectrum for $k$ is the same as the one for $-k$. Figure 7 shows the value of the fit coefficient $\alpha$ in each symmetry sector for $N = 20$ and $N = 21$. It is close to 0 in most cases, indicating Wigner behaviour. The same behaviour is observed for different values of $\kappa$ in $[-1, -0.1]$ and $[0.1, 1]$ — the unfolding method used isn't reliable for $|\kappa| < 0.1$ and $|\kappa| > 1$ since $n(E)$ becomes highly discontinuous near 0 and the difference between consecutive energies approaches the numerical precision. Therefore, our level spacing analysis suggests that the system isn't integrable for finite, non-zero $\kappa$. The value of $\alpha$ increases as $S_T^z$ increases because there are less eigenvalues so the histogram is less

smooth and the quality of the fit diminishes. The same behaviour is observed for $13 \leq N \leq 19$.

Our analysis allows us to conclude that the system is not integrable for generic value of $\kappa \neq 0, \pm\infty$. This applies equally to the Hamiltonian $H$ and the Markov operator $\hat{W}$, as they share the same spectrum. Our conclusion is in agreement with the known results for non-equilibrium classical Kawasaki dynamics [8].

## 5   Conclusion

To summarize, we have studied a stoquastic quantum spin-1/2 chain dual to the non-equilibrium Kawasaki dynamics of a classical Ising chain coupled to a thermal bath. After deriving the corresponding Hamiltonian for non-uniform Ising couplings, we showed that the exact groundstates are obtained from the Boltzmann distribution of the classical Ising model via a nonunitary similarity transformation given in Eq. (5). Energy level spacing distributions have revealed the model to be non-integrable for finite uniform couplings $\kappa \neq 0$. Consequently, we find that the associated non-equilibrium classical dynamics are ergodic. The one and two magnon sectors exhibit a critical exponent $z = 2$, which differs from the value $z = 3$ obtained when $\kappa \gg 1$, thus suggesting the presence of multiple dynamics at low energy. For the two-magnon sector, there is peculiar behavior in the antiferromagnetic regime past $\kappa_x = -\ln(3)/4$ as the nearest neighbor dominated branch crosses the NNN one, and starts penetrating the continuum. At $\beta = \infty$, the conservation of the Ising energy (i.e. domain wall number) leads to a multitude of frustrated states and the slowing down of the dynamics. Further work to understand the associated glassy dynamics is needed. It would also be interesting to generalize the analysis to the disordered case, as well as to higher dimensions.

From a quantum perspective, it would be interesting to examine the entanglement properties of the eigenstates. For example, one could study the quantum error correcting properties of its ground space, in the same spirit as was done for the Motzkin and other spin chains [21]. Further, given the conservation of spin, one could study the frequency-dependent spin conductivity as a function of the coupling $\kappa$, and of the temperature in the quantum system (not to be confused with the $1/\beta$ appearing in the coupling).

## Acknowledgments

The authors thank X. Chen, N. Crampé, H. Katsura, M. Knap and J. Feldmeier for insightful discussions. This work was funded by a Discovery Grant from NSERC, a Canada Research Chair, a grant from the Fondation Courtois, and a "Établissement de nouveaux chercheurs et de nouvelles chercheuses universitaires" grant from FRQNT. Simulations were performed on Calcul Québec's and Compute Canada's superclusters. GL is supported by a BESC M scholarship from NSERC and a B1X scholarship from FRQNT.

## A   Two-magnon continuum at large $|\kappa|$

It can be shown that the commutator of the Hamiltonian $H$ in Eq. (7) and the Ising energy $H_{\text{Ising}} = \sum_i \sigma_i^z \sigma_{i+1}^z$ gives

$$[H, H_{\text{Ising}}] = 1 - \tanh^2 \kappa, \tag{34}$$

which is zero in the limit where $\kappa \to \pm\infty$. At $\beta = \infty$, this new conservation law strongly constrains the dynamics. For the two-magnon subsector, the interaction between the NN bound

state and the continuum is also suppressed. In particular, we find that Eqs. (20,21) become

$$0 = Eg(1),\tag{35}$$

$$3g(2) - \cos(K)g(3) = Eg(2),\tag{36}$$

and thus we find a state $\psi$ with $F(\text{NN}, \psi) = 1$ and $E = 0$ at every value of $K$ in Fig. 8a. Similarly for $\kappa = -\infty$, $\delta_\kappa = 1$ and $\alpha_\kappa = -1$ and so Eqs. (20,21) now respectively become

$$2 = Eg(1),\tag{37}$$

$$g(2) - \cos(K)g(3) = Eg(2),\tag{38}$$

and we again find a state with $F(\text{NN}, \psi) = 1$ but with $E = 2$ for every value of $K$ in Fig. 8b. The NNN branch in Fig. 8b is the reflection of the one in Fig. 8a about the $E = 2$ line.

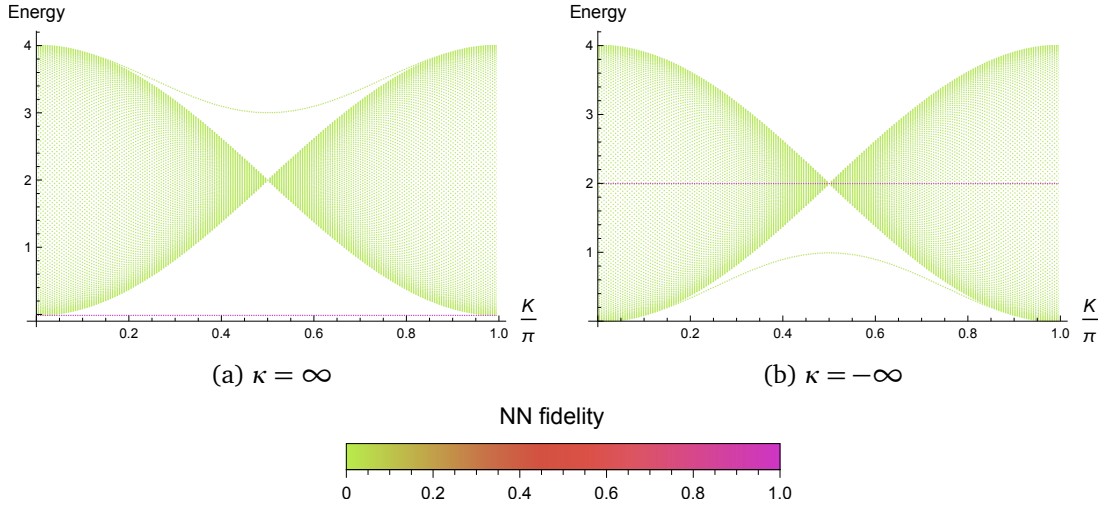

Figure 8: Two-magnon excitations and nearest neighbor fidelity for $N = 200$ and $\kappa = \pm\infty$ respectively in (a) and (b).

## B  Bloch basis

The common eigenstates of $S_T^z$ and the translation operator $T_1$ are the Bloch states which are given by

$$|\vec{s}, k_m\rangle = \frac{1}{\sqrt{p}} \sum_{n=0}^{p-1} e^{i n k_m} (T_1)^{-n} |\vec{s}\rangle,\tag{39}$$

for $\vec{s} \in \{-1, 1\}^N$ and $m \in \{0, 1, \ldots, N-1\}$ such that $p\, m/N \in \mathbb{N}$, where $k_m = 2\pi m/N$ is the momentum of the Bloch state and $p$ is the period of the state $|\vec{s}\rangle$, i.e. the smallest integer $p > 0$ such that $T_1^p |\vec{s}\rangle = |\vec{s}\rangle$.

## C  Numerical study of energy level spacings

Each state $|\vec{s}\rangle$ is stored as a binary number where each bit correspond to a site : 0 to ↓ and 1 to ↑. For each orbit of $T_1$ we select the state $\vec{s}$ with the lowest binary number as the representative

of its Bloch state $|\vec{s}, k\rangle$. To allow parallel computation of the Hamiltonian matrix elements and eigenvalues on a computer cluster, the basis states representatives are block-cyclically distributed using BLACS NUMROC function.

Because the action of the Pauli operators on the Bloch states are not straightforward, to compute the Hamiltonian elements in the Bloch basis we use the following simplification :

$$
\begin{aligned}
\langle \vec{s}', k | H | \vec{s}, k \rangle &= \left( \frac{1}{\sqrt{p'}} \sum_{n'=0}^{p'-1} e^{-in'k} \langle \vec{s}'| T^{\dagger}_{-n'} \right) H \left( \frac{1}{\sqrt{p}} \sum_{n=0}^{p-1} e^{ink} T_{-n} |\vec{s}\rangle \right) \\
&= \frac{1}{\sqrt{pp'}} \sum_{n'=0}^{p'-1} \sum_{n=0}^{p-1} e^{i(n-n')k} \langle \vec{s}'| T_{n'-n} H |\vec{s}\rangle \\
&= \frac{1}{\sqrt{pp'}} \sum_{n=1-p'}^{p-1} \min\left(p'+n, p-n, p', p\right) e^{ink} \langle \vec{s}'| T_{-n} H |\vec{s}\rangle \\
&= \frac{1}{\sqrt{pp'}} \sum_{i=1}^{N} \sum_{n=1-p'}^{p-1} \min\left(p'+n, p-n, p', p\right) e^{ink} \langle \vec{s}'| T_{-n} H_i |\vec{s}\rangle,
\end{aligned}
\tag{40}
$$

where $p$ and $p'$ are the periods of the representatives $\vec{s}$ and $\vec{s}'$, and

$$
H_i = \gamma_\kappa \left(1 + \delta_\kappa \sigma^z_{i-1} \sigma^z_{i+2}\right)\left(\sigma^x_i \sigma^x_{i+1} + \sigma^y_i \sigma^y_{i+1}\right) + \frac{1}{4}\left[1 + \alpha_\kappa \sigma^z_i \sigma^z_{i+2} - (1+\alpha_\kappa)\sigma^z_i \sigma^z_{i+1}\right].
\tag{41}
$$

To compute $\langle \vec{s}'| T_{-n} H_i |\vec{s}\rangle$ efficiently, bitwise operations on the binary numbers reprensenting $|\vec{s}\rangle$ and $|\vec{s}'\rangle$ are used.

Finally, all the eigenvalues of the block-cyclically distributed Hamiltonian matrix are found using the PZHEEV subroutine from ScaLAPACK.

## C.1 Finite-size scaling

As the chains get longer, the dimension of the sectors increases. Hence, the histogram of their level spacing distribution smoothens out and gets closer to the Wigner surmise. This can be seen in Fig. 9 where we see how $\alpha$ decreases with $N$. It suggests that the distribution tends to the Wigner surmise in the $N \to \infty$ limit.

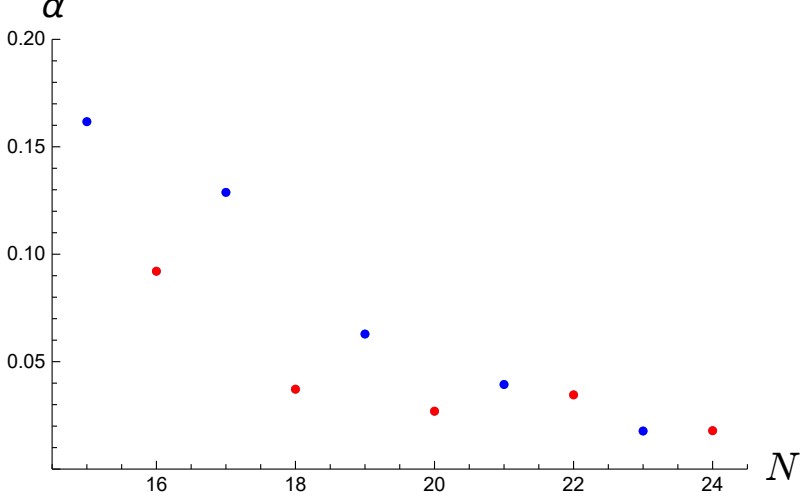

Figure 9: Fit coefficient $\alpha$ for increasing chain lengths $N$ in the lowest strictly positive $S^z_T$ and $k = 2\pi/N$ sectors. Blue dots have odd $N$ and $S^z_T = 1/2$. Red dots have even $N$ and $S^z_T = 1$.

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
