# Peer review of "Excitations and ergodicity of critical quantum spin chains from non-equilibrium classical dynamics"

_SciPost Physics Core, doi:SciPost Phys. Core 5, 035 (2022)_

## Round 2 · Referee Report · Anonymous (Referee 1) · 2022-1-30

Strengths

  • The model and analysis are mostly presented clearly.
  • The model appears to display some interesting phenomena such as glassiness and anomalous dynamics.

Weaknesses

  • Ultimately it is not clear what the significance of the results is.
  • The section on dynamical exponents is confusing. I am not sure what the result/claim is. The authors have access to the two-particle spectrum for all parameters so they should be able to settle the issue of how the excitation gap scales with size. Is the claim that the two-particle spectrum always has z = 2 but there could be other excitations that are slower? Could the value of kappa play any role here?

Report

I think this work does an interesting and apparently valid calculation.

Requested changes

Rewrite the section on dynamical critical exponents to make the claims more precise.

---

## Round 3 · Referee Report · Anonymous (Referee 1) · 2022-3-29

Report

The authors have addressed my concerns. I recommend publication at this stage.

---

## Round 3 · Author Response

We thank the referee for the report. Below we reply to the comments and questions.

1) “Ultimately it is not clear what the significance of the results is.”

Answer: One of the goals was to better understand Kawasaki dynamics in 1d using the arsenal of quantum spin chains. Among others, we uncovered highly non-trivial behaviour in the 2-magnon sector as a function of \kappa. This also leads to the identification of multiple critical dynamics, which were observed in other quantum spin chains of interest (Motzkin and Fredkin models). We also note that we provided the quantum numbers of the first two excited states. At \beta=\infty, we identified the exact degeneracy, which differs from the classical situation due to the singular nature of the transformation between the Markov operator and the quantum Hamiltonian. Our demonstration of ergodicity is in agreement with classical Monte Carlo results, but is more detailed since we have done this in numerous symmetry sectors.

2) “The section on dynamical exponents is confusing. I am not sure what the result/claim is. The authors have access to the two-particle spectrum for all parameters so they should be able to settle the issue of how the excitation gap scales with size. Is the claim that the two-particle spectrum always has z = 2 but there could be other excitations that are slower? Could the value of kappa play any role here?”

“Rewrite the section on dynamical critical exponents to make the claims more precise.”

Answer: In that section, we put our results on the 1- and 2-magnon sectors into context by comparing to what is known for all sectors. We conclude that the chain hosts multiple dynamics: diffusive modes (z=2), and at least one sub-diffusive one. \kappa indeed plays an important role, as a larger \kappa softens the sub-diffusive mode, while it leaves the diffusive ones unchanged. We have added numerous statements (in red in the new version) in order to clarify the discussion, and have added new results (like quantum numbers of excited states).

---

## Editorial Decision

published